# Mapping imported malaria in Bangladesh using parasite genetic and human mobility data

Hsiao-Han Chang[1,2†], Amy Wesolowski[3†], Ipsita Sinha[4,5], Christopher G Jacob[6], Ayesha Mahmud[1,2], Didar Uddin[4], Sazid Ibna Zaman[4], Md Amir Hossain[7], M Abul Faiz[4,8], Aniruddha Ghose[9], Abdullah Abu Sayeed[9], M Ridwanur Rahman[10], Akramul Islam[11], Mohammad Jahirul Karim[12], M Kamar Rezwan[13], Abul Khair Mohammad Shamsuzzaman[14], Sanya Tahmina Jhora[14], M M Aktaruzzaman[12], Eleanor Drury[6], Sonia Gonçalves[6], Mihir Kekre[6], Mehul Dhorda[4,5,15], Ranitha Vongpromek[15], Olivo Miotto[4,6,16], Kenth Engø-Monsen[17], Dominic Kwiatkowski[6,16], Richard J Maude[1,2,4,5†], Caroline Buckee[1,2†]*

[1]Department of Epidemiology, Harvard T.H. Chan School of Public Health, Boston, United States; [2]The Center for Communicable Disease Dynamics, Harvard T.H. Chan School of Public Health, Boston, United States; [3]Department of Epidemiology, Johns Hopkins Bloomberg School of Public Health, Baltimore, United States; [4]Mahidol-Oxford Tropical Medicine Research Unit, Faculty of Tropical Medicine, Mahidol University, Bangkok, Thailand; [5]Centre for Tropical Medicine and Global Health, Nuffield Department of Medicine, University of Oxford, Oxford, United Kingdom; [6]Wellcome Sanger Institute, Cambridge, United Kingdom; [7]Department of Medicine, Chittagong Medical College, Chittagong, Bangladesh; [8]Dev Care Foundation, Dhaka, Bangladesh; [9]Chittagong Medical College Hospital, Chittagong, Bangladesh; [10]Shaheed Suhrawardy Medical College, Dhaka, Bangladesh; [11]BRAC Centre, Dhaka, Bangladesh; [12]National Malaria Elimination Programme, Dhaka, Bangladesh; [13]Vector-Borne Disease Control, World Health Organization, Dhaka, Bangladesh; [14]Communicable Disease Control, Directorate General of Health Services, Dhaka, Bangladesh; [15]Worldwide Antimalarial Resistance Network, Asia Regional Centre, Bangkok, Thailand; [16]Big Data Institute, Oxford University, Oxford, United Kingdom; [17]Telenor Research, Telenor Group, Fornebu, Norway

*For correspondence: cbuckee@hsph.harvard.edu

†These authors contributed equally to this work

Competing interests: The authors declare that no competing interests exist.

**Abstract** For countries aiming for malaria elimination, travel of infected individuals between endemic areas undermines local interventions. Quantifying parasite importation has therefore become a priority for national control programs. We analyzed epidemiological surveillance data, travel surveys, parasite genetic data, and anonymized mobile phone data to measure the spatial spread of malaria parasites in southeast Bangladesh. We developed a genetic mixing index to estimate the likelihood of samples being local or imported from parasite genetic data and inferred the direction and intensity of parasite flow between locations using an epidemiological model integrating the travel survey and mobile phone calling data. Our approach indicates that, contrary to dogma, frequent mixing occurs in low transmission regions in the southwest, and elimination will require interventions in addition to reducing imported infections from forested regions. Unlike risk maps generated from clinical case counts alone, therefore, our approach distinguishes areas of frequent importation as well as high transmission.

DOI: https://doi.org/10.7554/eLife.43481.001

## Introduction

A global decline of malaria in recent decades has led to a push for complete national elimination of human malaria parasites in 21 countries by 2020 (*WHO, 2018*). In elimination and pre-elimination settings, malaria transmission is often highly heterogeneous geographically (*Bousema et al., 2012*; *Carter et al., 2000*; *Sturrock et al., 2016*), and policy-makers must focus on reducing transmission in remaining endemic foci, while protecting areas where malaria has been effectively controlled from imported infections that threaten to reintroduce parasites and reignite transmission (*Cotter et al., 2013*; *Churcher et al., 2014*). These tasks require different interventions, so understanding how patterns of regular travel to and from malaria endemic regions of a country contribute to the spread of malaria is a critical component of elimination planning. Measuring the spatial spread of malaria remains technically challenging, however, due to frequent asymptomatic and undetectable infections (*Lindblade et al., 2013*), as well as the inherent challenges associated with epidemiological surveillance in rural areas among highly mobile populations.

Decisions about how to allocate resources for malaria control and elimination are generally based on reports from hospitals, clinics, community health workers (CHWs) and non-governmental organizations (NGOs) around the country, which provide a measure of the incidence of symptomatic cases (*WHO, 2007*; *Moonen et al., 2010*). In areas with high mobility, however, the reporting health facility may not accurately capture local transmission, instead detecting imported infections that have been acquired elsewhere. Traditionally, patient travel surveys have been the principal method for identifying imported cases, but surveys can be unreliable and limited in scope, and importation cannot always be confirmed even when a travel history is accurate (*Wesolowski et al., 2014*). We and others have used spatially explicit epidemiological models– for example, parameterized using mobile phone data– to estimate 'sources and sinks' of parasites and the impact of travel on the spread of malaria (*Wesolowski et al., 2012*). These methods show promise, but do not provide highly spatially resolved estimates in regions with low cell tower density, such as in forested or sparsely populated regions, and may suffer from ownership biases.

There is also a growing interest in the routine use of sequencing to genotype parasites, which may provide insights into the spatial spread of malaria, particularly in low transmission settings where imported infections that sustain remaining hotspots must be addressed and in places where drug resistance poses a threat (*Amato et al., 2018*; *Busby et al., 2016*; *Imwong et al., 2017*; *Karunaweera et al., 2014*; *MalariaGEN Plasmodium falciparum Community Project et al., 2016*; *Miotto et al., 2015*; *Obaldia et al., 2015*; *Taylor et al., 2017*; *Tun et al., 2015*). However, in spite of decreasing costs, the use of genetic data is currently constrained by the lack of appropriate methods for analysis. *Plasmodium falciparum* infections are often composed of multiple different parasite clones, which undergo sexual recombination in the mosquito (*Mzilahowa et al., 2007*). As a result, phylogenetic and population genetic methods normally applied to the analysis of non-recombining organisms cannot be used. In addition to applying human population genetic tools to malaria parasites (*Chang et al., 2012*; *Lawson et al., 2012*; *Miotto et al., 2013*; *Patterson et al., 2006*; *Pritchard et al., 2000*), an emerging field of promising new approaches, such as THE REAL McCOIL, hmmIBD, or isoRelate, are being developed to explicitly address these characteristics of malaria infection (*Taylor et al., 2017*; *Chang et al., 2017*; *Henden et al., 2018*; *Schaffner et al., 2018*).

Conceptually, genetic and epidemiological approaches should complement each other – genetic data encodes information about the relatedness of parasites and mixing patterns between different parasite populations, while epidemiological data provides insights into clinical cases and the patterns of transmission. For viral pathogens like the influenza virus, where recombination does not obscure relationships between lineages, phylodynamic frameworks have been developed for inferring migration patterns and identifying the factors contributing to the migration rate (*Lemey et al., 2014*; *Nelson et al., 2015*; *De Maio et al., 2015*; *Vaughan et al., 2014*). Although it is possible to measure population differentiation on continental scales for malaria (*Miotto et al., 2015*), elimination programs often require inferences on smaller spatial scales, which are currently more challenging because of frequent recombination and the complex epidemiological dynamics of the parasite. In particular, on local spatial scales genetic signals may not be easily resolved because parasites are likely to be highly related, and comparisons between many pairs of populations become

computationally prohibitive (*De Maio et al., 2015*; *Vaughan et al., 2014*; *Rasmussen et al., 2014*). Developing methods that leverage insights from genomic data, epidemiological information, and mobility data sources such as travel survey or mobile phone data, is therefore an important goal for malaria, particularly if the collection of clinical samples for genetic analysis and travel data can be incorporated into routine surveillance workflows to support decision making for control programs.

The National Malaria Elimination Programme (NMEP) in Bangladesh currently estimates the burden of malaria using symptomatic case counts aggregated at administrative level 3 (*upazila*) or 4 (*union*). Typical of malaria in Southeast Asian countries, much of the transmission is thought to occur in forests and forest fringes where the principal vectors abound (*Bhatia et al., 2013*). The difficulties of surveillance in hard-to-reach forested areas complicate decisions about where to target transmission-reducing interventions (often focused on vector control) and how to find and treat imported cases as Bangladesh moves towards elimination. Moreover, Bangladesh – one of the countries that plan to eliminate malaria nationally – borders the Greater Mekong Subregion (GMS) (*WHO, 2017b*; *Haque et al., 2014*), where evidence of resistance to first line treatment for malaria (artemisinin combination therapy, or ACT) has been found in all countries (*WHO, 2017a*). Although resistance to artemisinin and to ACT partner drugs has not been found to date locally, Bangladesh is a potential point of passage in the spread of drug resistance out of the GMS, so monitoring drug resistance markers is crucial.

Epidemiological models of malaria designed to understand spatial transmission are often based on clinical incidence and seroprevalence estimates; data and methods required for integrating insights about importation from other data sources, including genetic data, are currently lacking (*Daniels et al., 2015*). Mobility patterns derived from mobile phone data are difficult to access, and rarely coincide temporally with epidemiological and genetic data, but could in theory provide real-time estimates of population movements. We collected and analyzed paired genetic, travel survey, and epidemiological malaria case data across the Chittagong Hill Tracts (CHT) region of Bangladesh, in combination with estimated population mobility patterns from mobile phone calling data. We developed a simple metric to quantify genetic mixing from parasite genetic barcode data. Geographic patterns of genetic mixing were then compared to estimates of malaria importation from a mathematical model, based on clinical epidemiological data and on mobility data from travel surveys and mobile phone records. We show that, taken together, these data sources provide evidence of heterogeneous transmission outside the high-incidence forested areas, and of substantial importation of parasites throughout the CHT. We propose that this type of data integration and analytical pipeline could help national control programs effectively target resources for malaria elimination.

## Results

The map of clinical incidence from the NMEP for 2015 and 2016 shown in *Figure 1A* in the CHT supports the hypothesis that transmission is highest in the eastern, forested region, similar to other countries in the GMS with forested border areas (*Figure 1B*). The location of the reporting health clinic may not reflect local acquisition of infection, however, particularly in regions like Bangladesh where laborers, forest-workers, and military personnel may travel frequently in and out of high transmission areas, but report to clinics in low transmission areas. We aimed to map remaining transmission foci in the CHT; in particular to identify areas outside forested regions where transmission is being sustained, rather than imported from the forest, since these areas should be targeted by transmission reduction interventions. To this end, we enrolled 2090 patients with confirmed malaria, attending 57 hospitals and clinics across the malaria-endemic region (see *Figure 1—figure supplement 1* for a map of sample distribution). All patients answered a travel questionnaire and their dried blood spots were collected on filter paper. In total, parasite genotypes of 1412 patients were obtained successfully. These data were combined with reported incidence for the region, and data on aggregated daily movement patterns from 2.4 million mobile phone subscribers over 6 months across the CHT (see Materials and methods).

### Disparate data sets identify broad division of transmission foci

We first analyzed each type of data separately, focusing on quantifying signals of spatial connectivity and on measuring transmission intensity (*Figure 1C,D,E*). Given the importance of distinguishing between the forested border regions – where most malaria is thought to occur – and less forested

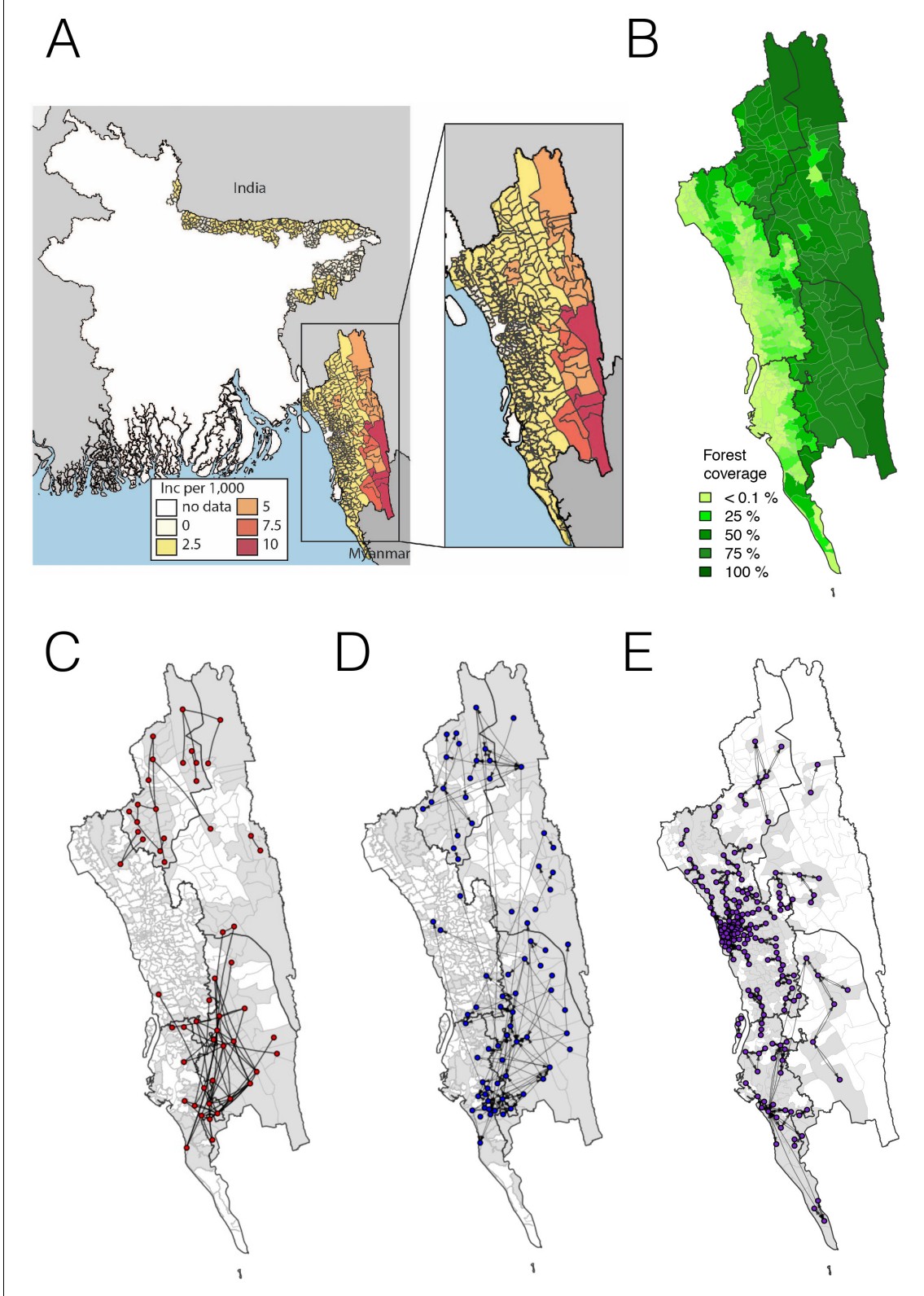

**Figure 1.** The incidence of malaria and spatial connectivity in the Chittagong Hill Tracts (CHT) Region. (A) The average monthly incidence per 1000 population of *P. falciparium* malaria from 2015 to 2016 as reported by the NMEP. Incidence was highest in the eastern portion of the CHT (shown in relation to the country borders) and decreased westward. (B) The forest coverage (%). (C) Unions sharing at least one parasite with identical genetic barcodes. (D) Top 50% of most traveled routes reported between pairs of locations from the travel survey data. (E) Top 1% of routes traveled between

*Figure 1 continued on next page*

*Figure 1 continued*

pairs of locations from the mobile phone data. Unions were colored grey where data was collected on genetic (**C**), travel survey (**D**) or mobile phone data (**E**).

DOI: https://doi.org/10.7554/eLife.43481.002

The following figure supplements are available for figure 1:

**Figure supplement 1.** Sample distribution.

DOI: https://doi.org/10.7554/eLife.43481.003

**Figure supplement 2.** Drug resistant markers and the proportion of identical parasites showing spatial signal.

DOI: https://doi.org/10.7554/eLife.43481.004

**Figure supplement 3.** Commonly used genetic measures show little spatial signal.

DOI: https://doi.org/10.7554/eLife.43481.005

areas elsewhere, for clarity we defined forested areas as those with >50% forest coverage. In the travel survey, 31% ($N = 654$) of malaria positive individuals reported living outside forested areas. Of these individuals, the majority did not report any travel to forested areas (66%, $N = 434$), suggesting that either individuals were acquiring infections outside these areas, or that the travel survey did not capture all forest travel. The survey indicated a general separation between the north and south portions of the endemic region, with fewer people traveling between these two regions. Within the regions, however, substantial travel was reported between the high incidence forested areas and the lower incidence ones (*Figure 1D*). Similarly, by simply connecting those pairs of unions where we identified parasites with identical genetic barcodes, a broad north-south division of the CHT region clearly emerged, as did connections between forested and non-forested areas (*Figure 1C*). Temporally, parasite pairs with identical genetic barcodes were significantly closer in time than those that were not identical (medians = 16.5 vs. 145 days, Mann-Whitney test p-value$<2\times10^{-16}$). Conversely, we found that other genetic diversity metrics, except for drug resistance-related markers, showed limited spatial resolution (see *Figure 1—figure supplements 2* and *3*). Roughly consistent with the north-south divide, mobility estimates derived from mobile phone calling data suggested a high amount of travel in the coastal areas around the cities of Chittagong and Cox's Bazar, with little travel between these areas (*Figure 1E*). This separation into two main transmission foci was consistent with the larger geographic distance between these areas of the CHT, but provided little insight into the extent of transmission heterogeneity on more local spatial scales.

To assess whether patterns of genetic relatedness among parasites showed signals consistent with survey- or mobile phone-based estimates of mobility, we analyzed all parasite pairs, along a continuum of genetic similarity, with respect to mobility outcomes. As expected, genetically similar parasites (fewer differences in genetic barcode comparisons) were more likely to come from the same residence unions (*Figure 2A*). For parasites from different unions, more related parasites were more likely to come from unions where direct travel had been reported in the survey (*Figure 2B*; see Materials and methods). To ensure this signal was not obscured by local effects, since both travel and genetic similarity is expected to be high between neighboring unions, we separated the data into unions less than 20 km apart and unions more than 20 km apart (*Figure 2—figure supplement 1*). The association shown in *Figure 2B* was mainly driven by unions that were $\geq$20 km apart, indicating that travel is responsible for closely related parasites in more geographically separated unions. Remarkably, closely related parasites that were neither from the same location *nor* from locations with direct travel reported were more likely to come from unions with indirect travel reported to a shared third location (*Figure 2C*). Using the travel survey data at the individual patient level instead of population level, we found that individuals infected with closely related parasites were significantly more likely to report living, working, or traveling to the same locations, confirming the correlation between the genetic and travel survey data (*Figure 2—figure supplement 2*). Comparing the genetic data to travel measured using mobile phone data, we found that genetically similar parasites that were not from the same locations were more likely to come from locations with greater inferred connectivity between them (*Figure 2D*), even though the majority of travel measured was in less forested areas. Therefore, although the travel survey and mobile phone data represent different geographic and population coverage, and disparate sample sizes (*Figure 2—figure supplement 3*), spatial signatures of connectivity were generally consistent with the genetic data.

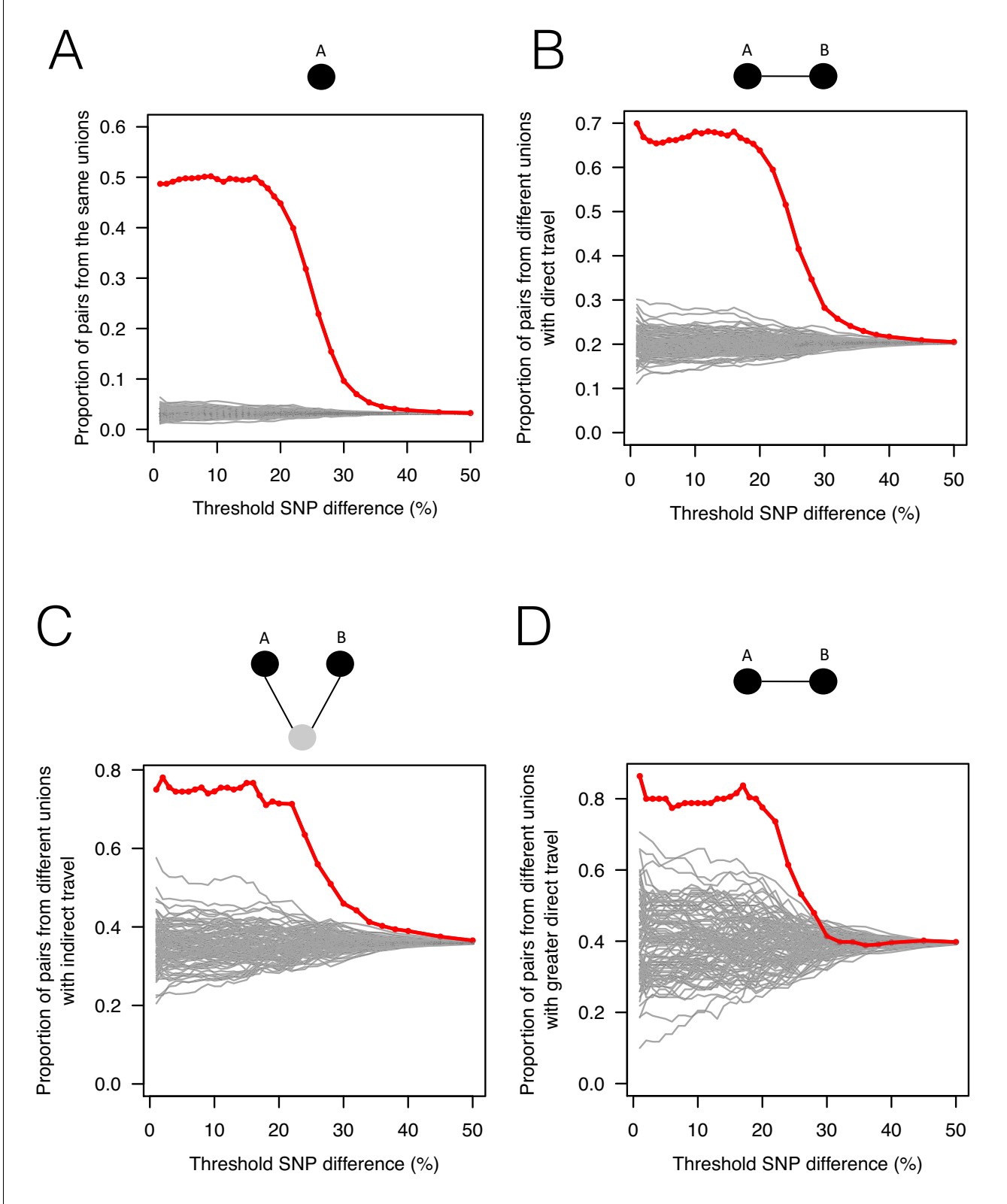

**Figure 2.** The association between genetic data, travel survey, and mobile phone data. We compared genetic data with population-level travel survey data and mobile phone data under four scenarios: being in the same union (**A**), coming from unions with direct travel reported in travel survey (**B**) or indirect travel (**C**), and coming from unions with high direct travel inferred from mobile phone data (**D**) (See Materials and methods for details). We calculated the proportion of pairs under these four scenarios when SNP differences were lower than given thresholds (red) and compared them with

*Figure 2 continued on next page*

*Figure 2 continued*

100 permutation results (grey). (**A**) Sample pairs with smaller SNP differences were more likely to be from the same union than random permutations; (**B**) if they did not live in the same union, they were more likely to be from unions with direct travel; (**C**) if neither of these conditions held, then they were more likely to be from unions with indirect travel. (**D**) If sample pairs were not from the same union, those with a smaller SNP difference were more likely to be from unions with higher direct travel (>0.1%) estimated from mobile phone data.

DOI: https://doi.org/10.7554/eLife.43481.006

The following figure supplements are available for figure 2:

**Figure supplement 1.** The association between SNP difference and travel at varying distances.

DOI: https://doi.org/10.7554/eLife.43481.007

**Figure supplement 2.** Odds ratio of observing nearly identical barcodes with respect to resident locations and travel patterns.

DOI: https://doi.org/10.7554/eLife.43481.008

**Figure supplement 3.** The differences in mobile phone versus travel survey data.

DOI: https://doi.org/10.7554/eLife.43481.009

## Genetic and model-based evidence of transmission in low incidence areas

To map transmission heterogeneities on smaller spatial scales, these rich data sources can be analyzed using more sophisticated approaches. We developed a metric designed to quantify the amount of parasite genetic mixing – indicative of frequent parasite importation and recombination between different parasites from different locations – in different regions. This index was calculated for each location with respect to all other locations as a risk ratio; specifically, the ratio of risks that parasites were genetically similar given that they were far away and nearby. To identify this index, we used the empirical relationship between genetic similarity and geographic distance for this region, so the thresholds defined would be setting-specific if rolled out routinely as part of surveillance programs (see Materials and methods). In brief, we quantified the decline of parasite genetic similarity with geographic distance, uncovering a robust signal; pairs of parasites sampled from unions that are geographically closer were more likely to be genetically similar (*Figure 3A*). In addition to the enrichment of genetically similar parasites in spatial proximity, there was a temporal signal: parasite pairs with SNP differences smaller than 10% were significantly closer in time than those with SNP differences greater than 10% (medians = 14 vs. 145 days, Mann-Whitney test p-value$<2\times10^{-16}$). Using the empirical relationship between geographic and genetic distances, we assumed a uniform prior on geographic distance and calculated the probability of geographic distances between patients given different levels of genetic similarity (*Figure 3—figure supplement 1*; see Materials and methods). This allowed us to define a CHT-specific 'genetic mixing index' to identify higher sharing of parasite barcodes between samples from relatively distant locations, indicative of high gene flow or importation (*Figure 3B* and *Figure 3—figure supplement 3*; see Materials and methods). *Figure 3B* maps this index across the CHT, showing heterogeneous patterns of mixing in the southwestern region, with high mixing regions indicating frequently imported infections, and low genetic mixing suggesting predominantly local transmission. In contrast, we did not observe elevated genetic mixing in the north (see robustness analysis in *Figure 3—figure supplement 2* and *Supplementary file 1* for 95% confidence intervals of the index).

Genetic measures based on SNP barcodes were able to identify unions with a high likelihood of importation events, but not the origin of imported infections since sharing genetic similarity did not provide information about the direction of movement. Epidemiological models that describe directional travel patterns may be used to combine reported incidence with data from travel surveys or mobile phone data to reconstruct routes of importation. We used mobility estimates from the survey and from mobile phone data to infer the amount of travel between locations in a simple epidemiological model that predicts the rate of infections moving between unions and mapped the resulting malaria source and sink locations (see Materials and methods, *Figure 4A,B*, and *Figure 4—figure supplements 1* and *2*). We compared the relative contribution of imported infections to local transmission in each union and predicted that locations in the southwestern area of the CHT had the highest proportion of imported cases, using both the travel survey and mobile phone data (*Figure 4A, B*). The mobility patterns derived from mobile phone data suggested more infections were being imported overall, however qualitatively both mobile phone data and travel surveys identified overlapping areas of the country where the top locations of importations were occurring. The genetic

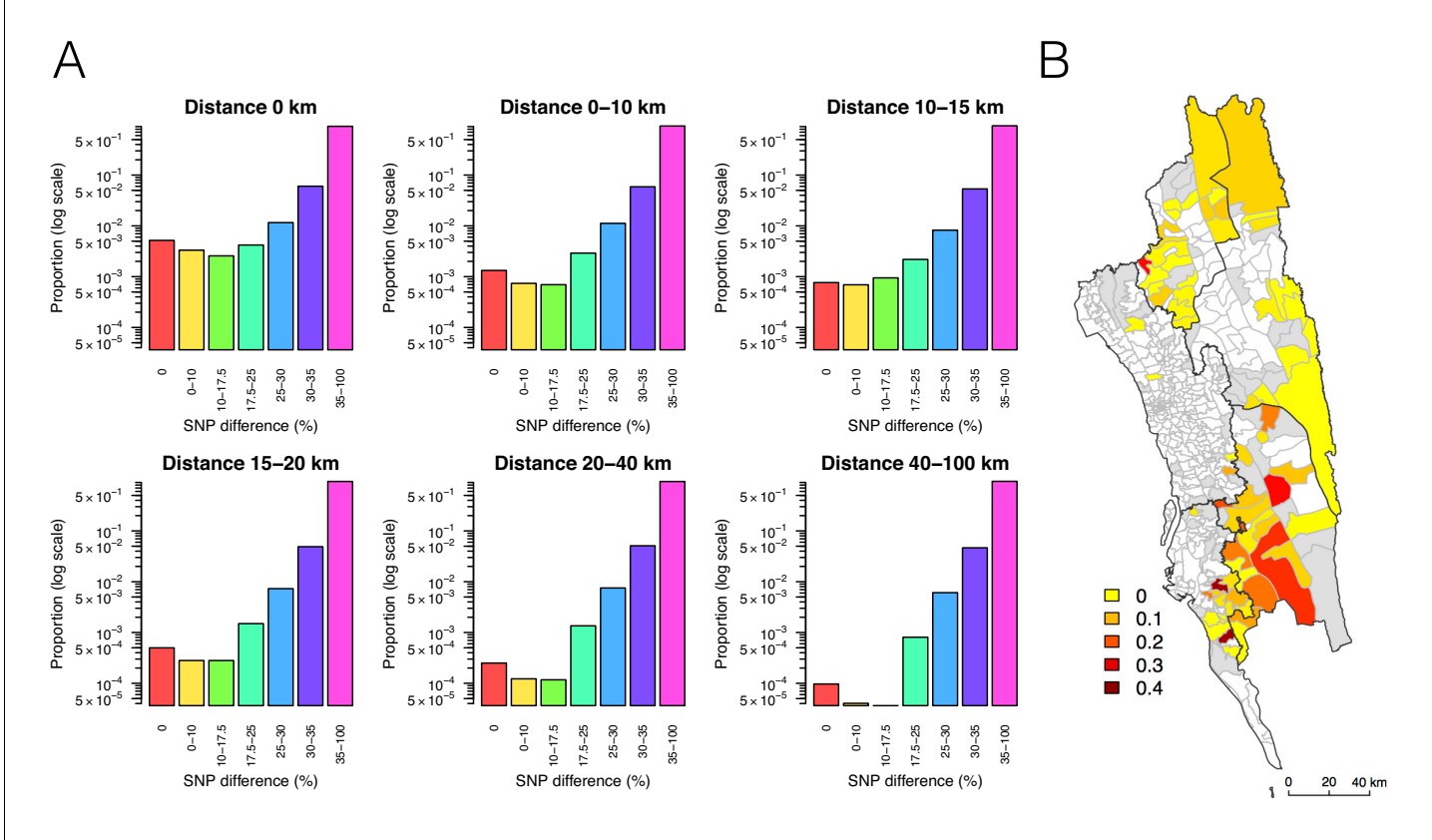

**Figure 3.** The relationship between genetic and geographic distance. (A) The association between genetic data and geographic distance was only obvious for small SNP differences. Pairs of parasites sampled from unions that are geographically closer were more likely to be genetically similar. The proportion of intermediate or high SNP differences did not vary much with geographic distance. (B) The genetic mixing index for each location. Unions were colored white if they did not include genetic data and grey if they included genetic data but their genetic mixing index was not identifiable due to lack of samples that were both nearby and genetically similar. High genetic mixing index suggests high parasite flow or importation.

DOI: https://doi.org/10.7554/eLife.43481.010

The following figure supplements are available for figure 3:

**Figure supplement 1.** The probability that parasites were sampled from locations within a specified geographic distance (red – purple) given different levels of SNP differences.

DOI: https://doi.org/10.7554/eLife.43481.011

**Figure supplement 2.** Genetic mixing index was robust to subsampling randomly and geographically.

DOI: https://doi.org/10.7554/eLife.43481.012

**Figure supplement 3.** Examples of genetic mixing index.

DOI: https://doi.org/10.7554/eLife.43481.013

mixing index therefore provides an independent confirmation that extensive parasite flow is occurring in the highly populated southwestern region of the CHT, while the epidemiological model – parameterized using travel survey data and mobile phone data – provides insights into the sources of imported infections.

## Generating combined transmission maps

Combining unions with a high genetic mixing index together with unions with a high proportion of imported cases estimated from the epidemiological models, we created an updated risk map for malaria transmission in the CHT (*Figure 5D*). Both the genetic mixing index and the epidemiological models suggest frequent mixing in the southwest of the CHT, in Cox's Bazar district (*Figures 3B* and *4A,B*). The epidemiological model predictions were roughly consistent with the unions exhibiting a high genetic mixing index (*Figure 4C*). Additionally, we identified frequent importation *within* the southwest region from epidemiological models parameterized using both travel survey and mobile

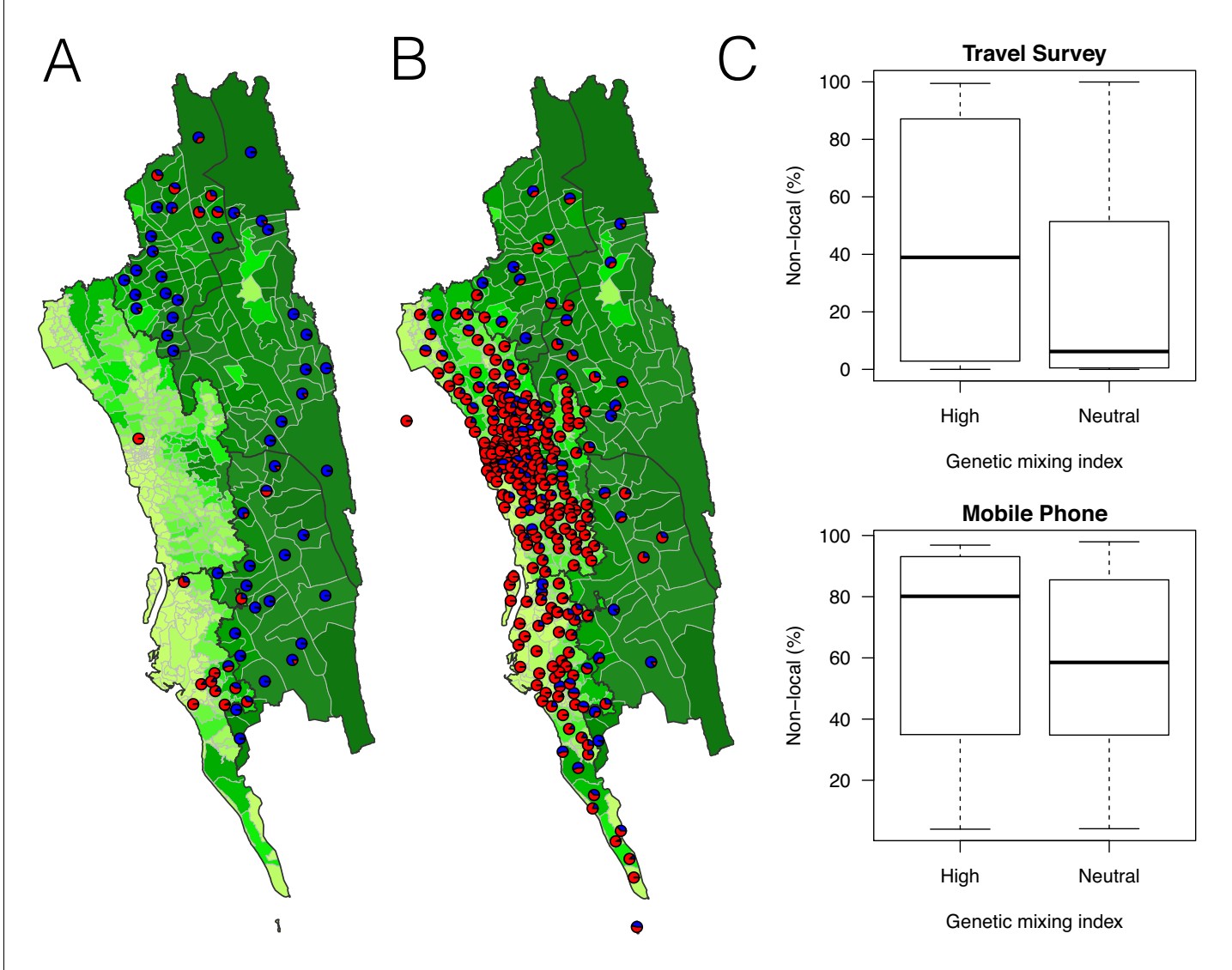

**Figure 4.** The estimated non-locally acquired cases from the epidemiological modeling using travel survey and mobile phone data. (A–B) The estimated proportions of non-locally acquired parasites. We estimated the percentage of infections in each union that were acquired in other destination unions (red) versus locally acquired (blue). For each union where travel data was available from the travel survey (**A**) or mobile phone data (**B**), the percentage was shown. Unions were colored according to their forest coverage (light green for low, dark green for high). (**C**) We classified genetic mixing index >0.1 as high and ≤0.1 as neutral. The estimated proportion of imported infections from the travel survey data (top panel) or mobile phone data (bottom panel) was higher for unions classified as a high genetic mixing index, while not statistically significant (p-value>0.05).
DOI: https://doi.org/10.7554/eLife.43481.014

The following figure supplements are available for figure 4:

**Figure supplement 1.** The estimated non-locally acquired cases from the epidemiological modeling using travel survey and mobile phone data.
DOI: https://doi.org/10.7554/eLife.43481.015

**Figure supplement 2.** Sources of parasite importations based on the epidemiological models parameterized by the mobility data.
DOI: https://doi.org/10.7554/eLife.43481.016

phone data. In the model parameterized using travel survey data, parasites were predicted to be frequently imported from the forested areas of the CHT to the more populated, lower transmission areas, particularly unions within Cox's Bazar district (*Figure 5A*, *Figure 5—figure supplements 1* and *2*). The model parameterized by mobile phone data also identified the forested areas as a source, and in addition, suggested that parasites were imported from more populated, lower transmission areas in the west (*Figure 5B*). Since travel surveys and mobile data had quite different

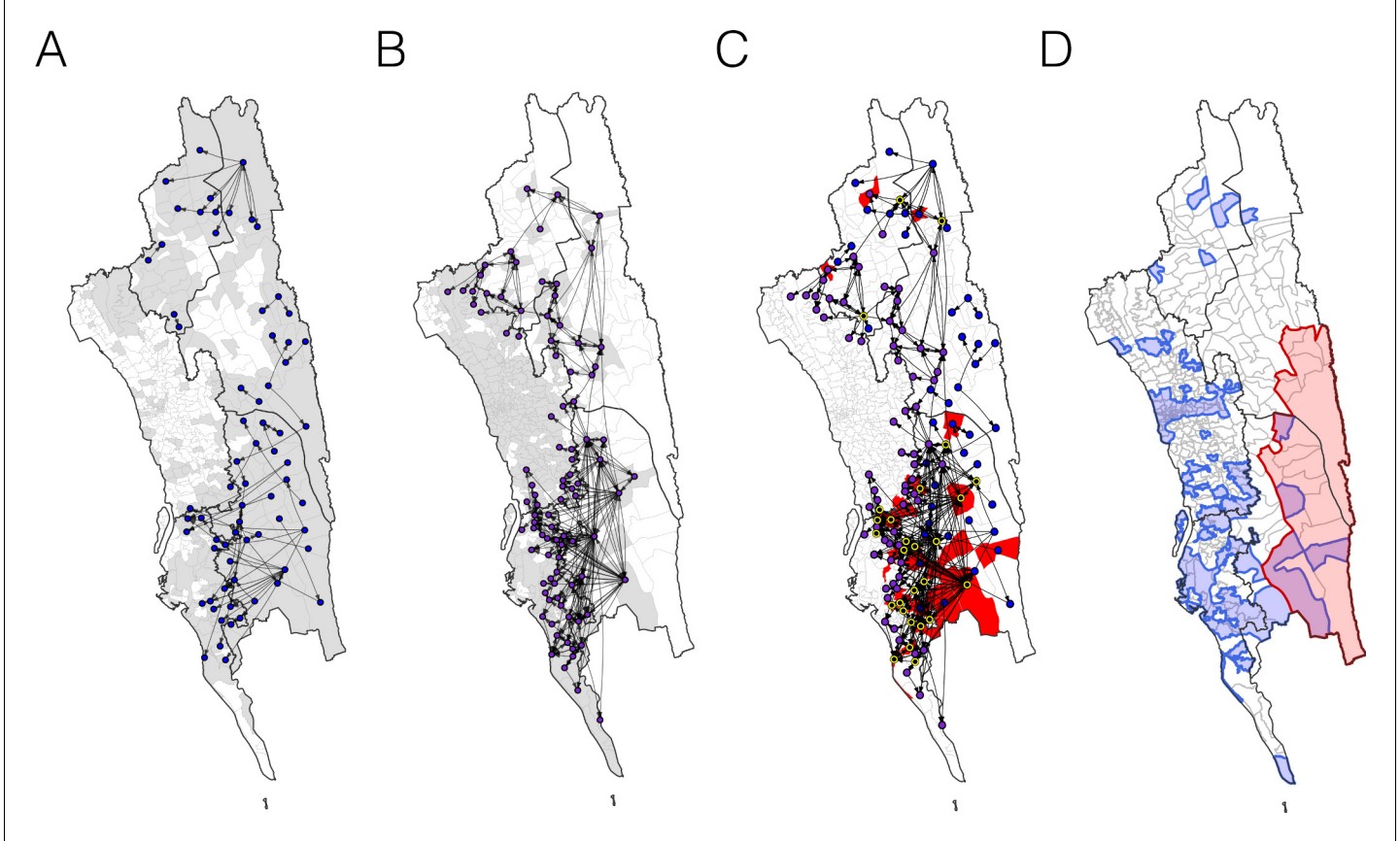

**Figure 5.** The estimated routes of parasite importations. (A–B) We estimated parasite flows between unions using epidemiological models parameterized by the travel survey (A) or mobile phone data (B). The top routes of parasite flows (the top 25% for travel survey and the top 1% for mobile phone) and their origins and destinations were shown by lines and dots, respectively. Unions were colored grey if they included travel survey (A) or mobile phone data (B). The top routes of parasite flows accounted for 86.4% (travel survey) and 87.8% (mobile phone) of the total importation. (C) The combined map showing top parasite importation routes from the travel survey (nodes colored blue), mobile phone data (nodes colored purple), or both (nodes colored black with a yellow outline). Unions were shown in red if they had a high genetic mixing index. (D) An updated risk map for malaria transmission in the CHT: unions with high genetic mixing index (>0.1) or a high proportion of imported cases estimated from the epidemiological models (blue: top 25% from the travel survey, top 1% from the mobile phone data), high incidence areas (red: the average monthly incidence per 1000 population >4), and unions that have both high incidence and a high importation level (purple).

DOI: https://doi.org/10.7554/eLife.43481.017

The following figure supplements are available for figure 5:

**Figure supplement 1.** The top 10 routes of parasite flows.

DOI: https://doi.org/10.7554/eLife.43481.018

**Figure supplement 2.** The top routes of importation based on different types of travel.

DOI: https://doi.org/10.7554/eLife.43481.019

geographic coverage and different biases (*Figure 1C,D*), some discrepancies are expected. Nevertheless, these analyses support the ideas that a) forested areas are an important source of imported infections, consistent with current dogma, but b) human movement around densely populated regions with relatively low transmission appears to contribute substantially to heterogeneous transmission across the southwest of the CHT (*Figure 5C*). This means that resources for elimination should be targeted in these areas, in addition to the forested regions. The updated map (*Figure 5D*) highlights the regions which were not identified in the incidence map (*Figure 1A*), for example. Our results also suggest that importation routes occur separately in the north and south of the Chittagong region (*Figure 5C* and *Figure 5—figure supplement 1*), implying that a staged elimination strategy could be effective.

## Discussion

Routine surveillance for malaria among control programs is primarily based on prevalence estimates and/or clinical incidence reported by hospitals, clinics, community health workers and non-governmental organizations. The geographic distribution of clinical cases may not reflect patterns of transmission, particularly in areas hosting highly mobile populations, where importation of parasites is common. Identifying the true foci of transmission is particularly important in elimination settings, such as Bangladesh. A number of approaches have been suggested to augment risk maps based on incidence data, including molecular surveillance and epidemiological models parameterized by new data streams, such as mobile phone data. Here, our goal was to assess whether a combination of these approaches could provide actionable insights for control programs. As a complement to a simple map of clinical incidence, or model-based geospatial maps based on incidence and prevalence estimates (https://map.ox.ac.uk), this combined approach illustrated that, while the eastern forested regions are indeed contributing imported infections to the lower transmission areas in the southwest, substantial local importation and genetic mixing is also occurring in these highly populated areas along the southwestern part of the CHT. This suggests that, while targeting interventions to forested areas is a key strategy for elimination, it might not be successful unless the mixing between low transmission settings in the southwest region is also addressed. We propose that, with appropriate sampling strategies, parasite genetic analysis could provide actionable insights for national control programs and help evaluate the success of elimination programs.

Our samples came from symptomatic patients who presented at a health facility, which would be the norm for routine reporting to the NMEP. We therefore did not include asymptomatic or subclinical infections, and it is unknown to what extent the parasite in symptomatic patients are representative of the entire parasite population in this area (*Searle et al., 2017*). Although we were able to enroll patients at 57 health facilities that covered a wide geographic area, these study sites only represent 59% of all CHT health facilities which report data to the NMEP. In several health facility catchment areas, non-government organizations provide malaria treatment, testing, and education outside health facilities as part of a malaria outreach initiative. In these areas, we may be under-estimating true incidence, since fewer people seek malaria treatment at the government health clinics. Although these biases may impact epidemiological modeling results, the genetic mixing index – which may lack power when sample size is small, but is not biased by underestimating incidence – can provide complementary insights.

In this transmission setting and geography, genetic differentiation based on genetic barcodes between parasite populations cannot be easily distinguished by commonly used methods, such as average pairwise difference or $F_{ST}$ (*Figure 1—figure supplement 3*). Consistent with our previous studies showing that the proportion of nearly identical genetic sequences is more sensitive to recent migration of pathogens (*Taylor et al., 2017*; *Chang et al., 2016*), in our data the proportion of parasites with nearly identical genetic barcodes was highly associated with geographic distance. This allowed us to develop a genetic mixing index, which may provide context-specific information about gene flow. However, the thresholds used for calculating this index would need to be determined in each setting independently, because the association between geographic distance and genetic similarity can vary between settings. This index could also be used at the individual level, to identify infections that are likely to be imported. We found three cases with a genetic mixing index greater than 1, suggesting that they were likely to be imported and, consistently, there was reported travel between their residence unions and the unions they shared similar barcodes with. The reliance of the genetic mixing index and other spatial approaches on nearly-identical parasites – which are rare, but hold the most meaningful genetic signals – means that small sample sizes and large uncertainty are inevitable challenges to their routine use. Although increasing sample size would improve our estimates, adding more SNPs to the analysis is unlikely to reduce uncertainty of the index, since the patterns of relatedness are an intrinsic feature of malaria epidemiology in this region. Nevertheless, in combination with other modeling approaches, genetic data can yield insights that are not attainable any other way – particularly in the context of elimination planning. As molecular surveillance becomes more widespread, clearly establishing the strengths and limitations of genetic data will be critical.

Mobile phone data and travel surveys are different from each other in terms of scale, geographic coverage, and granularity of data about human mobility in this context (*Wesolowski et al., 2018*).

Our travel survey provides detailed, individual-level information about infected individuals, although its quality is dependent on the respondent's ability to accurately recall their travel over the past two months. Although we included specific questions to help with recall bias, such as asking explicitly about travel to the forested areas, these biases likely still exist and may have limited our ability to infer parasite movement. Overall, travel survey data were sparse at the union level, and many individuals reported no travel (287 out of 2,090), or only travel within a single union (1240 out of 2,090). In contrast, mobile phones provide large volumes of data about the connectivity of the general population, but no individual-level information, and only in locations where cell towers exist. The contrasting geographic coverage and scale of the two sources of data on human mobility – mobile phone and travel survey data – lead to different estimates of parasite importation. Mobile phone data provides insights into large-scale population flows, but is limited to regions with cell towers, while travel survey data provide detailed information about a small number of malaria-infected individuals. Both data sources are biased, but together can provide complementary insights into the population dynamics underlying the spread of parasites in the region. Given the difficulties associated with measuring population mobility and the importance of understanding importation in countries aiming for elimination, multiple (albeit biased) estimates of human travel provide the most complete picture of parasite flows. Given the biases in the incidence data (discussed above), in the epidemiological model, we chose a simple approach without having to estimate the underlying transmission or directly incorporating vectorial capacity. Remarkably, travel patterns from both data sources were consistent with results from parasite genetic analyses, confirming their complementarity.

Every epidemiological and genetic data set has inherent limitations and biases, but multiple data layers can complement each other effectively to provide insights into different components of malaria transmission. This study represents a first example of the utility of combining these sources of information in the context of identifying imported infections, and we believe it shows that cost-effective sequencing approaches can be combined with simple epidemiological models to provide a more complete picture of malaria transmission on a subnational level.

## Materials and methods

### Malaria epidemiological and mobility data

Patients self-presenting to 58 health facilities across hill districts Bandarban, Khagrachari, and Rangamati and adjacent coastal districts Chittagong, and Cox's Bazar in Bangladesh from January 2015 to September 2016 and tested positive for malaria were recruited (*Figure 1—figure supplement 1*). In total, 2090 individuals were included in the travel survey. The majority of these individuals were infected with *P. falciparium* ($N = 1,540$), followed by *P. vivax* ($N = 332$), mixed infections ($N = 215$), and unknown ($N = 3$). Trips from an individual's residence village were quantified for work, frequent, infrequent, forest, and international travel including the destination (geocoded to the union level, 176 unions in total were included in the analysis), number of nights spent on the trip, frequency of travel (number of trips), and timing (when did the trip occur). Travel survey questions are listed in *Supplementary file 2*. We also analyzed mobile phone calling data to estimate population-level mobility within the CHT. Mobile phone call data records were analyzed from 1 April – 30 September 2017 where individuals were assigned their most frequently used mobile phone tower per day. Tower locations were aggregated to the corresponding union. Travel between unions was calculated when subscribers' most frequently used tower location (aggregated to the union) changed between consecutive days. Subscribers who did not change their locations were assumed to have remained in the same location on these days. Over this time period, we calculated the average number of daily trips between unions within the CHT using previously developed methods (*Wesolowski et al., 2015*). Over half of the unions included in the travel survey included at least one mobile phone tower ($N = 147$, 64%), however coverage was limited in the eastern most unions of the CHT.

The total number of confirmed (by microscopy and RDT) *Plasmodium falciparum* cases is reported monthly to the NMEP (*Laskar et al., 2018*). These data were aggregated to the corresponding upazila ($N = 95$) per month from January 2015 to August 2018, and were used to calculate the average monthly malaria incidence using the estimated population sizes from the most recent census (*Bangladesh Bureau of Statistics, 2011*). For a subset of unions ($N = 141$), monthly incidence data

was available for unions and was used in lieu of the upazila level incidence data (*Figure 1A*). The percent of forest coverage was calculated using estimates by Hansen et al. (*Hansen et al., 2013*). Geographic distances were calculated as road distance between union centroids using R package 'Geosphere' (*Karney, 2013*).

## Genetic analysis

We performed barcode assays on parasite samples from all of the 2090 malaria positive patients enrolled, and successfully obtained genetic barcodes of parasite samples from 1412 individuals who resided in 134 separate unions. Barcodes were formed from genotypes at 101 SNPs at locations across the genome known to be highly differentiating. These SNPs are variable in all major global geographic regions and have moderate minor allele frequencies in these populations, as well as above average levels of between population $F_{ST}$. Genotypes were produced using the mass-spectrometry based platform from Agena (*Supplementary file 4* and *6*). In addition, samples were genotyped for known drug resistance-related markers (*Supplementary file 3*). We estimated complexity of infection and population allele frequencies using THE REAL McCOIL (*Chang et al., 2017*) and included both monoclonal and polyclonal samples in the analysis. The percentage of SNP difference was calculated using the number of SNP differences divided by the total number of sites excluding missing data and assuming that SNP difference between heterozygous call and homozygous call is 0.5.

We defined a genetic mixing index as follows:

$$\text{Genetic mixing index}_i = \frac{\text{Prob}\left(\text{genetically similar} \mid \text{far}\right)}{\text{Prob}\left(\text{genetically similar} \mid \text{close}\right)} = \frac{\frac{n_{f,s}}{n_{f,s}+n_{f,ns}}}{\frac{n_{c,s}}{n_{c,s}+n_{c,ns}}}.$$

For each sample from a particular union *i* we classified all other samples as either genetically similar or dissimilar, and calculated the number of these samples that were geographically 'far' or 'close'. The numbers were then aggregated across individuals within each union *i*: $n_{f,s}$, $n_{f,ns}$, $n_{c,s}$, and $n_{c,ns}$ represent the number of samples that are far (*f*) or close (*c*) and genetically similar (*s*) or not genetically similar (*ns*). Since it is more likely to observe between individuals who have physically nearby residences, if a sample had a higher-than-expected probability of sharing genetic similarity with samples far away, it is likely that it was imported. We used samples that were 'not genetically similar' to control for the effect of sample sizes. This index is expected to increase with the proportion of imported cases and the number of 'sources' (see examples in *Figure 3—figure supplement 3*). The 95% confidence intervals were calculated as follows (*Katz et al., 1978*):

$$Exp\left(log(genetic\ mixing\ index) \pm 1.96\sqrt{\frac{1}{n_{f,s}} - \frac{1}{n_{f,s}+n_{f,ns}} + \frac{1}{n_{c,s}} - \frac{1}{n_{c,s}+n_{c,ns}}}\right).$$

In this study, $\leq 17.5\%$ SNP difference was chosen as the threshold for genetic similarity and $\geq 20$ km was chosen as the threshold for geographically far because the empirical probability of having $\leq 17.5\%$ SNP difference above 20 km was smaller than 0.05; the threshold of >30% SNP difference was used to determine 'not genetically similar' parasites because the probability of observing >30% SNP difference varied minimally with geographic distance in Bangladesh (*Figure 3—figure supplement 1*). Genetic mixing index >0.1 and $\leq 0.1$ were classified as 'high' and 'neutral', respectively.

## The association between genetic data, travel survey, and mobile phone data

We compared genetic data with both population-level and individual-level travel survey data. At the population level, we examined how SNP differences (denoted by *x*) related to the travel survey data and compared the empirical results with 100 permutations (*Figure 2*). We considered three scenarios: (1) individuals living in the same union (denoted by $T_1$), (2) individuals coming from places with direct travel (denoted by $T_2$), and (3) individuals coming from places with indirect travel (denoted by $T_3$). Two unions were considered to have 'indirect' travel if they were both connected by travel to another union that had non-zero incidence. Specifically, we calculated the proportion of parasite pairs under these three scenarios, given different SNP thresholds (denoted by *S*), as follows:

$$Prop\,(T_1\,|x<S) = \frac{Prop\,(T_1,\,x<S)}{Prop\,(x<S)}$$

$$Prop\,(T_2\,|x<S,\,not\,T_1) = \frac{Prop\,(T_2,\,x<S\,|\,not\,T_1)}{Prop\,(x<S\,|\,not\,T_1)}$$

$$Prop\,(T_3\,|x<S,\,not\,T_1,\,not\,T_2) = \frac{Prop\,(T_3,\,x<S\,|\,not\,T_1,\,not\,T_2)}{Prop\,(x<S\,|not\,T_1,not\,T_2)}$$

We excluded $T_1$ when calculating $T_2$, and excluded $T_1$ and $T_2$ when calculating $T_3$, in order to identify the signal associated with each scenario separately. Our results show that parasite pairs with smaller SNP differences were more likely to come from the same unions, unions with direct, or unions with indirect travel, than random permutations (**Figure 2A,B,C**), indicating that genetic data was consistent with travel survey.

We performed a similar analysis using mobile phone data. Because almost all pairs of locations have direct travel inferred from mobile phone data, instead of calculating the proportion of locations with direct travel, we calculated the proportion of parasite pairs from locations with *higher* direct travel (>0.1%). The results show parasite pairs with smaller SNP differences, not living in the same unions, were more likely to come from unions with higher direct travel, indicating the association between genetic similarity and mobile phone data (**Figure 2D**).

At the individual level, we calculated the odds ratio of observing nearly identical barcodes with respect to the residence location or travel pattern was calculated as follows:

$$Odds\,ratio = \frac{\frac{Prob(nearly\,identical\,barcoes|\,C)}{Prob(not\,nearly\,identical\,barcodes|\,C)}}{\frac{Prob(nearly\,identical\,barcodes|\,not\,C)}{Prob(not\,nearly\,identical\,barcodes|\,not\,C)}}$$

where $C$ is the residence location or travel pattern (e.g., the condition that two individuals live in the same union, or work in the same union, or travel to the same union, etc.). Nearly identical barcodes were defined as barcodes with less than a 10% SNP difference, and 'not nearly identical barcodes' was defined as barcodes with SNP differences between the 25th and 75th percentiles for all SNP differences. Based on the estimated population allele frequencies, expected pairwise similarity between two random parents is 0.558 and therefore the expected parent-offspring similarity after one outcrossing event is 0.779 (=0.5 + 0.5*0.558). This indicates that the SNP difference thresholds of 10% and 17.5% used here are conservative, falling within one outcrossing event (i.e. lower than 1–0.779 = 22.1%) and represent high similarity and recent ancestor sharing.

## Quantifying the probability of a geographic distance given a SNP difference

To investigate the relationship between geographic and SNP distance, we considered seven geographic distance windows $D$= (0, 0–10, 10–15, 15–20, 20–40, 40–100, >100 km) and six SNP difference windows $S$= (0, 0–10, 10–17.5, 17.5–25, 25–30, 30–35%). To establish what genetic distance alone tells us about the spatial relationships between isolates, we assumed a uniform prior with geographic distance. The inferred geographic distances (based on genetic data) can then be compared to the true geographic distance between samples collected by the control program, to assess whether isolates are imported or not. For all pairs of unions within a specified geographic distance window $d$, the proportion of sample pairs with SNP differences within each SNP difference window $s$, $Prob(s|d)$, was calculated. We then calculated the probability of a geographic distance $d$ given a SNP difference $s$, $Prob(d|s)$, by applying Bayes' rule and assuming a uniform prior as follows:

$$Prob(d|s) = \frac{Prob(d,s)}{Prob(s)} = \frac{Prob(s|d)\,Prob(d)}{\sum_{\text{all}\,d\in D}Prob(s|d)Prob(d)} \overset{\text{uniform prior}}{\Rightarrow} \frac{Prob(s|d)}{\sum_{\text{all}\,d\in D}Prob(s|d)}.$$

Here, a uniform prior allows the genetic data alone to speak; depending on the question of interest, more realistic priors can be incorporated by weighting based on the known sampling distances.

## Modeling parasite flow using travel survey and mobile phone data

We calculated the number of trips between all pairs of unions from either the travel survey or mobile phone data. For the travel survey data, all individuals self-reported a residence union. We aggregated the trips to all other destinations by individuals for each residence union. For individuals who did not report any travel, we assumed they remained in their residence union. Based on the design of the travel survey, individuals could report travel for four separate reasons from their residence location (work, forest, frequent, and infrequent trips) (*Supplementary file 2*). For individuals who reported travel for fewer options, we assumed that a non-report was equivalent to remaining in their residence union (i.e. if they only reported a travel destination for work, then for the remaining travel questions – forest, frequent, and infrequent – we assumed that the individual was at their residence location). We further normalized the number of trips between unions by the total number of trips originating from each residence union had, and obtained the proportion of trips from union $i$ to union $j$ for all pairs of unions, $T_{ij}$, where $\sum_j T_{ij} = 1$ for each $i$.

Among 176 unions which had travel survey data, 104 of them (59%) had at least one mobile phone tower. We calculated the number of trips between unions with mobile phone towers between consecutive days and used the average number of trips over 6 months. Similar to the travel survey, $T_{ij}$ was calculated by normalizing the number of trips each union made.

## Epidemiological estimates of parasite importations

We estimated the parasite flow based on travel from the residence union scaled by the relative incidence in the destination union versus the residence union (see *Supplementary file 5* for travel matrices inferred from travel survey and mobile phone data). The proportion of infections in union $i$ coming from union $j$, $P_{ij}$, was calculated as follows:

$$P_{ij} = \frac{T_{ij}\frac{incidence_j}{incidence_i}}{\sum_j T_{ij}\frac{incidence_j}{incidence_i}} = \frac{T_{ij}\, incidence_j}{\sum_j T_{ij}\, incidence_j}$$

$P_{ii}$ and $(1-P_{ii})$ were the estimated proportion of local transmission and importations, respectively. We calculated a population-level measure of parasite flow, the number of cases in union $i$ from union $j$ ($M_{ij}$), based on the number of clinical cases in each residence union as follows:

$$M_{ij} = (\text{Population size in union } i)\,(incidence_i)P_{ij}$$

## Acknowledgments

We thank Joseph Lewnard for helpful discussion on statistical analysis, and the staff of Wellcome Sanger Institute Sample Management, Genotyping, Sequencing and Informatics teams for their contribution.

## Additional information

### Funding

| Funder | Grant reference number | Author |
|---|---|---|
| National Institute of General Medical Sciences | U54GM088558 | Hsiao-Han Chang |
| Burroughs Wellcome Fund | | Amy Wesolowski |
| Bill and Melinda Gates Foundation | CPT000390 | Ipsita Sinha Sazid Ibna Zaman Richard J Maude |
| Bill and Melinda Gates Foundation | OPP1129596 | Ipsita Sinha Sazid Ibna Zaman Richard J Maude |

| Medical Research Council | G0600718 | Christopher G Jacob<br>Eleanor Drury<br>Sonia Gonçalves<br>Mihir Kekre<br>Dominic Kwiatkowski |
| --- | --- | --- |
| Bill and Melinda Gates Foundation | OPP1118166 | Christopher G Jacob<br>Olivo Miotto<br>Dominic Kwiatkowski |
| National Institute of General Medical Sciences | R35GM124715-02 | Caroline Buckee |

The funders had no role in study design, data collection and interpretation, or the decision to submit the work for publication.

## Author contributions

Hsiao-Han Chang, Amy Wesolowski, Formal analysis, Investigation, Methodology, Writing—original draft, Writing—review and editing; Ipsita Sinha, Kenth Engø-Monsen, Data curation, Formal analysis, Writing—review and editing; Christopher G Jacob, Didar Uddin, Sazid Ibna Zaman, Md Amir Hossain, M Abul Faiz, Aniruddha Ghose, Abdullah Abu Sayeed, M Ridwanur Rahman, Akramul Islam, Mohammad Jahirul Karim, M Kamar Rezwan, Abul Khair Mohammad Shamsuzzaman, Sanya Tahmina Jhora, M M Aktaruzzaman, Eleanor Drury, Sonia Gonçalves, Mihir Kekre, Mehul Dhorda, Ranitha Vongpromek, Olivo Miotto, Data curation, Writing—review and editing; Ayesha Mahmud, Formal analysis, Writing—review and editing; Dominic Kwiatkowski, Data curation, Supervision, Funding acquisition, Writing—review and editing; Richard J Maude, Conceptualization, Data curation, Supervision, Project administration, Writing—review and editing; Caroline Buckee, Conceptualization, Supervision, Funding acquisition, Investigation, Writing—original draft, Project administration, Writing—review and editing

## Author ORCIDs

Hsiao-Han Chang http://orcid.org/0000-0001-8016-1530
Amy Wesolowski http://orcid.org/0000-0001-6320-3575
Ipsita Sinha http://orcid.org/0000-0002-6574-310X
Kenth Engø-Monsen http://orcid.org/0000-0003-1618-7597
Caroline Buckee http://orcid.org/0000-0002-8386-5899

## Decision letter and Author response

Decision letter https://doi.org/10.7554/eLife.43481.028
Author response https://doi.org/10.7554/eLife.43481.029

# Additional files

## Supplementary files

• Supplementary file 1. Genetic mixing index.
DOI: https://doi.org/10.7554/eLife.43481.020

• Supplementary file 2. Questions in the travel survey.
DOI: https://doi.org/10.7554/eLife.43481.021

• Supplementary file 3. Drug resistance markers.
DOI: https://doi.org/10.7554/eLife.43481.022

• Supplementary file 4. Genetic barcode data.
DOI: https://doi.org/10.7554/eLife.43481.023

• Supplementary file 5. Travel matrices inferred from travel survey and mobile phone data.
DOI: https://doi.org/10.7554/eLife.43481.024

• Supplementary file 6. Genetic barcode locations.
DOI: https://doi.org/10.7554/eLife.43481.025

• Transparent reporting form

DOI: https://doi.org/10.7554/eLife.43481.026

**Data availability**

All genetic data are included in Supplementary file 4 and all travel matrices are included in Supplementary file 5.

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
