## [Decision Letter]

Thank you for submitting your article "The geography of malaria elimination in Bangladesh: combining data layers to estimate the spatial spread of parasites" for consideration by *eLife*. Your article has been reviewed by three peer reviewers, and the evaluation has been overseen by Neil Ferguson as the Senior Editor and Reviewing Editor. The following individuals involved in review of your submission have agreed to reveal their identity: Edward A. Wenger (Reviewer #1); Oliver Brady (Reviewer #3).

The reviewers have discussed the reviews with one another and the Reviewing Editor has drafted this decision to help you prepare a revised submission.

The authors provide a logical framework for combining multiple data sources relating to travel history and parasite genetics, resulting in a single updated risk map for the Chittagong Hill Tracts region of Bangladesh. This updated map is informative of source-sink dynamics beyond simple prevalence/incidence measures produced by traditional epidemiological approaches and is therefore likely to be useful to NMEPs. Combining data layers in this way is a non-trivial problem due to the idiosyncrasies of different types of travel data and the complex signal in Plasmodium genetic data, and so this represents a significant step forward in terms of demonstrating the applied utility of Plasmodium genetic data over large spatial scales.

The manuscript is clear and well-written. Nevertheless, we have some comments that should be addressed:

First, this paper would benefit from more context in the presentation and interpretation of genetic signatures:

a) How do complex infections or inferred within-sample diversity vary among samples and between geographic regions? Does this tell a complementary story to the parasite pairwise relatedness?

b) Does presenting the relatedness information as a graph structure rather than bulk properties of pairwise relatedness provide any additional insights?

c) Can the authors give more context for choices of relatedness thresholds – 10% "nearly identical" (subsection “The association between genetic data, travel survey, and mobile phone data” last paragraph) and 17.5% similarity (subsection “Genetic analysis”, last paragraph). How do these relate to e.g. 1 or 2 outcrossing events with random parents drawn from population-level minor allele frequencies?

The big difference in coverage of travel survey and mobile phone data makes the assessment of their comparability / complementariness quite difficult to assess as a reader. While we appreciate there is not much the authors can do to change this, there are changes to the way the data are presented that could improve this, e.g. Figure 4 would be easier to interpret differences between the two data sources if only areas for which both travel survey and mobile phone data were available. The discussion could also elaborate more on the implications of this disparity in coverage.

Given the huge amount of work that has gone into this analysis, the final risk map (Figure 5D) is disappointingly non-specific. To what extent is this east-west flow of parasites a new result, or is it already known to some degree? If resources were limited, what would be the suggestion about how best to use them? Can top importation routes be predicted, and how much of total importation would they account for?

We have some concerns about the new genetic mixing index. This is presented as an odds ratio on samples being in near vs. far spatial unions given their level of genetic similarity, but does this not have the "exposure" and the "outcome" the wrong way around, i.e. what we observe is genetic similarity as a function of distance, not the converse? The symmetry property of odds ratios means that numerically swapping these round (i.e. Prob (genetically similar | far)) would result in the same final value, but would seem to make more intuitive sense.

There are many ways of constructing a statistic to measure the relationship between geographic and genetic distance, and so some statistical justification is needed for the use of an odds ratio. I'm sure the authors are aware that odds ratios are typically avoided in general due to frequent misinterpretation as risk ratios and their extreme behaviour in some parts of the parameter space (a small difference in probabilities could lead to a very small or very large difference in odds). For example, if the "rare disease" assumption is being used to justify this statistic then this should be stated. If no justification can be given then perhaps a risk-ratio would be more appropriate.

Some representation of uncertainty in the genetic mixing index should be given, either analytically or by simulation. Extreme uncertainty is another known weakness of odds ratios, hence it needs to be addressed.

There is a tendency to swap between talking about genetic similarity as a function of distance (e.g. subsection “Genetic and model-based evidence of transmission in low incidence areas”) and distance as a function of genetic similarity (e.g. Figure 3A) without appreciating that this involves a transformation. For example, when talking about the probability of geographic distance between parasite pairs (Figure 3) there must have been an application of Bayes' rule with a particular prior on distance, but this is not formally stated. Note that using the raw counts without a prior is equivalent to assuming a uniform prior on distance, which is almost certainly not appropriate here (we would not expect points to be uniformly separated a priori).

How sensitive are values of the genetic mixing index to non-uniform sampling of clinical infections or genetic sequences from different transmission regions or subpopulations?

Note – we understand and appreciate the intent of the genetic mixing index and feel it could be a valuable addition to the analytical toolbox, but it does need to pass certain basic statistical checks to avoid issues later on.

*Reviewer #1:*

This an important body of work, combining different sources of mobility data – travel surveys, mobile-phone records, parasite genetics – to quantify epidemiological and programmatic endpoints.

1) This paper would benefit from significantly more context in the presentation and interpretation of genetic signatures.

a) How do complex infections or inferred within-sample diversity vary among samples and between geographic regions? Does this tell a complementary story to the parasite pairwise relatedness?

b) Does presenting the relatedness information as a graph structure rather than bulk properties of pairwise relatedness provide any additional insights?

c) Can the authors give more context for choices of relatedness thresholds – 10% "nearly identical" (subsection “The association between genetic data, travel survey, and mobile phone data” last paragraph) and 17.5% similarity (subsection “Genetic analysis”, last paragraph). How do these relate to e.g. 1 or 2 outcrossing events with random parents drawn from population-level minor allele frequencies?

2) How sensitive are mixing-metric values to non-uniform sampling of clinical infections or genetic sequences from different transmission regions or subpopulations?

*Reviewer #2:*

The authors provide a logical framework for combining multiple data sources relating to travel history and parasite genetics, resulting in a single updated risk map for the Chittagong Hill Tracts region of Bangladesh. This updated map is informative of source-sink dynamics beyond simple prevalence/incidence measures produced by traditional epidemiological approaches, and is therefore likely to be useful to NMEPs. Combining data layers in this way is a non-trivial problem due to the idiosyncrasies of different types of travel data, and the complex signal in Plasmodium genetic data which is just starting to be leveraged effectively. This manuscript therefore represents a significant step forward in terms of demonstrating the applied utility of Plasmodium genetic data over large spatial scales.

One potential point of weakness in the analysis is the presentation of the new genetic mixing index. This statistic is presented as an odds ratio on samples being in near vs. far spatial unions given their level of genetic similarity. First, does this not have the "exposure" and the "outcome" the wrong way round, i.e. what we observe is genetic similarity as a function of distance, not the converse? The symmetry property of odds ratios means that numerically swapping these round (i.e. Prob (genetically similar | far)) would result in the same calculation, but would seem to make more intuitive sense.

Second, there are many possible ways of constructing a statistic to measure the relationship between geographic and genetic distance, and so some statistical justification is needed for the use of an odds ratio. I'm sure the authors are aware that odds ratios are typically avoided in general due to frequent misinterpretation as risk ratios and their extreme behaviour in some parts of the parameter space (a small difference in probabilities could lead to a very small or very large difference in odds). For example, if the "rare disease" assumption is being used to justify this statistic then this should be stated. If no justification can be given then perhaps a risk-ratio would be more appropriate. Some representation of uncertainty should also be given, as again this is a weakness of odds ratios. Keep in mind that this statistic may well be taken up by the community and applied/interpreted in other scenarios, and therefore it is important to ensure statistical robustness at this stage.

On a similar note, there is a tendency to swap between talking about genetic similarity as a function of distance (e.g. subsection “Genetic and model-based evidence of transmission in low incidence areas”) and distance as a function of genetic similarity (e.g. Figure 3A) without appreciating that this involves a transformation. For example, when talking about the probability of geographic distance between parasite pairs (Figure 3) there must have been an application of Bayes' rule with a particular prior on distance to obtain this. Note that using the raw counts without a prior is equivalent to assuming a uniform prior on distance, which is almost certainly not appropriate here.

I could not find a reference or SI detailing the 101 SNP barcode used. Are all these SNPs in linkage equilibrium? Given that other investigators are likely to apply the genetic mixing index on larger data sets, some comment on the effect of LD on this statistic would be welcome.

Other than these statistical issues I have no major concerns with the manuscript, which is well-written and makes a significant contribution to the field.

*Reviewer #3:*

This manuscript by Chang et al. details an interesting assessment of the combination of travel and genetic data to assess malaria elimination strategy in Bangladesh. It is highly novel, timely and interesting as well as being well described and reproducible. I have only a few comments on what I found an interesting, if a little difficult to follow, analysis.

In the first part of the results it is shown that human movement is important for defining the network of parasite similarity. However in the second part, the authors use a genetic similarity index that only uses simple geographic distance. I understand that the authors want to propose a easily calculatable index, but it would be nice to see the best risk map possible for Bangladesh given the data available.

The big difference in coverage of travel survey and mobile phone data makes the assessment of their comparability / complementariness quite difficult to assess as a reader. While I appreciate there is not much the authors can do to change this, there are changes to the way the data are presented that could improve this, e.g. Figure 4 – would be easier to interpret differences between the two data sources if only areas for which both travel survey and mobile phone data were available. The discussion could also elaborate more on the implications of this disparity in coverage.

Given the huge amount of through work that has gone into this analysis, the final risk map (Figure 5D) is disappointingly non-specific. If resources were limited, what would be the suggestion about how best to use them? Can top importation routes be predicted, and how much of total importation would they account for?

[Editors' note: further revisions were requested prior to acceptance, as described below.]

Thank you for resubmitting your work entitled "Mapping imported malaria in Bangladesh using parasite genetic and human mobility data" for further consideration at *eLife*. Your revised article has been favorably evaluated by Neil Ferguson as the Senior and Reviewing Editor, and three reviewers.

The manuscript has been improved but two reviewers require some remaining issues to be addressed before acceptance, as outlined below.

I would also concur with reviewer 2 that the SNP positions need to be included in this paper or a citation given to a published source. A preprint on bioRxiv would suffice. However, if a preprint is not yet published, please include the positions here.

*Reviewer #1:*

Thanks to the authors for the welcome additions and responses to reviewer feedback.

A reiteration of the substantial concern that publishing SNP positions with genetic sequences is an integral part of the work, especially in a journal dedicated to "improv[ing] research communication through open science and open technology innovation".

*Reviewer #2:*

I am happy that the new version does a good job of addressing the initial concerns, and I appreciate the work done by the authors.

My only remaining concern is with the application of Bayes rule to obtain Prob (distance | similarity). First, I agree with the choice to swap Figures 3 and Figure 3—figure supplement 1. But the main text reads that there was "no prior" on geographic distance – there was a prior, it was uniform as stated in the Materials and methods section. It is impossible to apply Bayes rule without the use of a prior, and a uniform prior does not represent no information, it actually represents quite specific information. If the intention is to let the genetic data "speak" and not worry about the prior probability of edges being a certain distance then perhaps it would be better to report this as a raw likelihood, i.e. Prob (similarity | distance). Alternatively if the authors really want to stick with the inverted probability then this could be smoothed over with a statement to the effect of "a uniform prior allows the genetic data alone to speak, and more realistic priors can be incorporated by weighting based on the known sampling distances". At the moment it is unfortunately misleading, because the probability of a randomly chosen pair of samples being a given distance apart based on their genetic similarity almost certainly *is not* given by Figure 3—figure supplement 1.

*Reviewer #3:*

The reviewers have sufficiently addressed all my comments in this revision

---

## [Author Response]

[…] The manuscript is clear and well-written. Nevertheless, we have some comments that should be addressed:First, this paper would benefit from more context in the presentation and interpretation of genetic signatures:a) How do complex infections or inferred within-sample diversity vary among samples and between geographic regions? Does this tell a complementary story to the parasite pairwise relatedness?

With the number of possible genetic analyses, we could have conducted and the space limit, we decided to constrain the number of genetic results we presented. However, we agree that we should present the COI results, since these were indicative of different transmission settings in other contexts (for example, Uganda [Chang et al., 2017]). We have now expanded the Results to show additional genetic results.

In particular, we now include a supplementary figure (Figure 1—figure supplement 3C) that shows the variation in the complexity of infection between geographic regions. In general, the complexity of infection was low (we did not observe an average COI greater than 2 in any union) and the differences between locations were small, and therefore did not provide additional insights into malaria transmission.

We mention that genetic measures shown in Figure 1—figure supplement 3 provided no or very limited spatial resolution in the Results as follows:

“Conversely, we found that other genetic diversity metrics, except for drug resistance-related markers, showed limited spatial resolution (see Figure 1—figure supplement 2 and 3).”

b) Does presenting the relatedness information as a graph structure rather than bulk properties of pairwise relatedness provide any additional insights?

In Figure 1—figure supplement 3E, we showed that average pairwise difference is not clearly associated with geographic distance. To compare with the graph structure shown in Figure 1C, we added one additional supplementary figure (Figure 1—figure supplement 3B) to show the average pairwise difference in the map, which did not show any obvious geographic pattern. These patterns of relatedness across these populations mean that the most informative signals are found among highly related parasites rather than in average pairwise relatedness in our study, and this likely also applies to proximate populations in other low transmission settings as well.

c) Can the authors give more context for choices of relatedness thresholds – 10% "nearly identical" (subsection “The association between genetic data, travel survey, and mobile phone data” last paragraph) and 17.5% similarity (subsection “Genetic analysis”, last paragraph). How do these relate to e.g. 1 or 2 outcrossing events with random parents drawn from population-level minor allele frequencies?

The expected similarity based on population-level allele frequencies is 55.8%, and therefore one outcrossing event corresponds to 77.9% parent-offspring similarity (0.779= 0.5+(1-0.5)*0.558), which is equivalent to 22.1% difference. This means the SNP difference thresholds of 10% and 17.5% used in our study were within one outcrossing events and represent high similarity and recent shared ancestors. We added this information to give more context for these thresholds in the Methods and Materials as follows:

“Based on the estimated population allele frequencies, expected pairwise similarity between two random parents is 0.558 and therefore the expected parent-offspring similarity after one outcrossing event is 0.779 (= 0.5+0.5*0.558). This indicates that the SNP difference thresholds of 10% and 17.5% used here are conservative, falling within one outcrossing event (i.e. lower than 1–0.779=22.1%) and represent high similarity and recent ancestor sharing.”

The big difference in coverage of travel survey and mobile phone data makes the assessment of their comparability / complementariness quite difficult to assess as a reader. While we appreciate there is not much the authors can do to change this, there are changes to the way the data are presented that could improve this, e.g. Figure 4 would be easier to interpret differences between the two data sources if only areas for which both travel survey and mobile phone data were available. The discussion could also elaborate more on the implications of this disparity in coverage.

We have added a new figure where only areas with both travel survey and mobile phone data were plotted (Figure 4—figure supplement 1). We also now discuss the implications of the disparity in coverage in the Discussion as follows:

“The contrasting geographic coverage and scale of the two sources of data on human mobility – mobile phone and travel survey data – lead to different estimates of parasite importation. […] Given the difficulties associated with measuring population mobility and the importance of understanding importation in countries aiming for elimination, multiple (albeit biased) estimates of human travel provide the most complete picture of parasite flows.”

Given the huge amount of work that has gone into this analysis, the final risk map (Figure 5D) is disappointingly non-specific. To what extent is this east-west flow of parasites a new result, or is it already known to some degree? If resources were limited, what would be the suggestion about how best to use them? Can top importation routes be predicted, and how much of total importation would they account for?

The dogma in Bangladesh is that nearly all malaria originates in the forest. While it is true that incidence is highest in the forest regions, this analysis highlights the importance of spread to the west, and the high volume of human travel around these regions. Our analysis suggests that the sheer volume of travel around the other lower transmission regions in the western part of Chittagong is likely to be sustaining transmission, and that these areas may be easier to target in any case. In terms of resource allocation, our study suggests that surveillance and control in lower transmission regions in southwestern Chittagong may be worthwhile, but that the importation routes that are most important are separate from north to south – therefore, it should be possible to conduct a staged elimination program focusing on each region separately. We have added the following section to the Results:

“This means that resources for elimination should be targeted in these areas, in addition to the forested regions. […] Our results also suggest that importation routes occur separately in the north and south of the Chittagong region, implying that a staged elimination strategy could be effective.”

With regards to the top importation routes: these are plotted in Figure 5 ABC and Figure 5—figure supplement 1. The total importation top routes account for is 86.4% and 87.8% for the epidemiological models parameterized by the travel survey and mobile phone data respectively. We have added the amount of total importation that the top importation routes account for in the legends of Figure 5 and Figure 5—figure supplement 1 as follows:

Figure 5:

“The top routes of parasite flows (the top 25% for travel survey and the top 1% for mobile phone) and their origins and destinations were shown by lines and dots, respectively. The top routes of parasite flows account for 86.4% (travel survey) and 87.8% (mobile phone) of the total importation.”

Figure 5—figure supplement 1:

“The top 10 routes of parasite flows using epidemiological models parameterized by the (A) travel survey (blue) or (B) mobile phone data (purple). These routes accounted for 39.1% (travel survey) and 40.2% (mobile phone) of the total importation.”

We have some concerns about the new genetic mixing index. This is presented as an odds ratio on samples being in near vs. far spatial unions given their level of genetic similarity, but does this not have the "exposure" and the "outcome" the wrong way around, i.e. what we observe is genetic similarity as a function of distance, not the converse? The symmetry property of odds ratios means that numerically swapping these round (i.e. Prob (genetically similar | far)) would result in the same final value, but would seem to make more intuitive sense.

After reading reviewers comments, we agree, and have decided to define genetic mixing index that is similar to a risk ratio rather than an odds ratio. We have revised the formula, and it is now Prob (genetically similar | far distance). Please see the responses to the comments below for more details.

There are many ways of constructing a statistic to measure the relationship between geographic and genetic distance, and so some statistical justification is needed for the use of an odds ratio. I'm sure the authors are aware that odds ratios are typically avoided in general due to frequent misinterpretation as risk ratios and their extreme behaviour in some parts of the parameter space (a small difference in probabilities could lead to a very small or very large difference in odds). For example, if the "rare disease" assumption is being used to justify this statistic then this should be stated. If no justification can be given then perhaps a risk-ratio would be more appropriate.

We did make the “rare disease” assumption because the chance of observing genetically similar pairs is very low. We compared our original genetic mixing index with risk ratios and the values are nearly identical (the largest difference is 0.00133 and the average difference is 0.00019), but we agree with the reviewer that risk-ratio would be more appropriate and therefore modify our definition of genetic mixing index in the Materials and methods as follows:

“

Genetic mixing indexi=Probgenetically similar | farProbgenetically similar | close=nf,snf,s+nf,nsnc,snc,s+nc,ns.

[…] The numbers were then aggregated across individuals within each union *i: n_f,s_, n_f,ns_, n_c,s_*, and *n_c,ns_* represent the number of samples that are far (*f*) or close (*c*) and genetically similar (*s*) or not genetically similar (*ns*).”

Some representation of uncertainty in the genetic mixing index should be given, either analytically or by simulation. Extreme uncertainty is another known weakness of odds ratios, hence it needs to be addressed.

We agree that uncertainty is an important factor. We have now reported 95% confidence intervals of genetic mixing index in Supplementary file 1 and mentioned this in the Results as follows:

“Figure 3B maps this index across the CHT, showing heterogeneous patterns of mixing in the southwestern region, with high mixing regions indicating frequently imported infections, and low genetic mixing suggesting predominantly local transmission. In contrast, we did not observe elevated genetic mixing in the north (see robustness analysis in Figure 3—figure supplement 2 and Supplementary file 1 for 95% confidence intervals of the index).”

The formula used to obtain confidence intervals was described in the Materials and methods as follows:

“The 95% confidence intervals were calculated as follows (Katz et al., 1978):

Explog(geneticmixingindex) ± 1.961nf,s-1nf,s+nf,ns+1nc,s-1nc,s+nc,ns.”

We emphasized that the uncertainty of genetic mixing index needs to be noted in the Discussion as follows:

“The reliance of the genetic mixing index and other spatial approaches on nearly-identical parasites – which are rare, but hold the most meaningful genetic signals – means that small sample sizes and large uncertainty are inevitable challenges to their routine use. […] As molecular surveillance becomes more widespread, clearly establishing the strengths and limitations of genetic data will be critical.”

There is a tendency to swap between talking about genetic similarity as a function of distance (e.g. subsection “Genetic and model-based evidence of transmission in low incidence areas”) and distance as a function of genetic similarity (e.g. Figure 3A) without appreciating that this involves a transformation. For example, when talking about the probability of geographic distance between parasite pairs (Figure 3) there must have been an application of Bayes' rule with a particular prior on distance, but this is not formally stated. Note that using the raw counts without a prior is equivalent to assuming a uniform prior on distance, which is almost certainly not appropriate here (we would not expect points to be uniformly separated a priori).

We agree with the reviewer that for most readers it is more intuitive to think about the probabilities of genetic similarity given geographic distance, and decided to replace Figure 3A with the original Figure 3—figure supplement 1 and move Figure 3A to a supplementary figure. We further include the following methodological updates below.

We chose a uniform prior since we wanted to show what genetic distance alone tells us without knowing where samples are collected, and to explicitly compare this with the “true” geographic distance. In particular, we developed the method to provide information about future samples as they are collected by the control program, who will want to have some idea whether the transmission event was local or imported. When the SNP difference between samples is small, these isolates are likely to be from nearby locations (Figure 3—figure supplement 1) or, if the true geographic distance is large, it is likely a non-local transmission event. On the other hand, because the probability of a large SNP difference does not vary much with geographic distance (Figure 3A), when the observed SNP difference is large (in our case >30%), we cannot infer anything about importation (all the colors in the sixth column of the figure have similar proportions). We have included an explanation in the Materials and methods as follows:

“To establish what genetic distance alone tells us about the spatial relationships between isolates, we assume a uniform prior with geographic distance. The inferred geographic distances (based on genetic data) can then be compared to the true geographic distance between samples collected by the control program, to assess whether isolates are imported or not.”

We have modified the text to make it clear that we assumed uniform prior as follows:

“Using the empirical relationship between geographic and genetic distances, we assumed no prior on geographic distance and calculated the probability of geographic distances between patients given different levels of genetic similarity (Figure 3—figure supplement 1; see Materials and methods).”

The details on how we calculated the probability of geographic distance using Bayes’ rule was in the Materials and methods as follows:

“Quantifying the probability of a geographic distance given a SNP difference

To investigate the relationship between geographic and SNP distance, we considered seven geographic distance windows *D*= (0, 0–10, 10–15, 15–20, 20–40, 40–100, >100 km) and six SNP difference windows *S*= (0, 0–10, 10–17.5, 17.5–25, 25–30, 30–35%). […] We then calculated the probability of a geographic distance *d* given a SNP difference *s,*Probds, by applying Bayes' rule and assuming uniform prior as follows:

Probds=Prob(d,s)Prob(s)=ProbsdProb(d)∑alld∈DProbsdProb(d)⇒uniform priorProbsd∑alld∈DProbsd.”

How sensitive are values of the genetic mixing index to non-uniform sampling of clinical infections or genetic sequences from different transmission regions or subpopulations?

The genetic mixing index is a “relative probability” based on the proportion of genetically similar parasite pairs between places that are geographically close or far. Because the expected value of proportion does not vary with sample size, genetic mixing index is not expected to be sensitive to non-uniform sampling from different transmission settings. However, for subpopulations with smaller sample sizes, the confidence interval of the genetic mixing index is expected to be higher or the index may not be identifiable. We performed subsampling and showed in Figure 3—figure supplement 2that the genetic mixing index remained qualitatively similar.

[Editors' note: further revisions were requested prior to acceptance, as described below.]

Thank you for resubmitting your work entitled "Mapping imported malaria in Bangladesh using parasite genetic and human mobility data" for further consideration at eLife. Your revised article has been favorably evaluated by Neil Ferguson as the Senior and Reviewing Editor, and three reviewers.The manuscript has been improved but two reviewers require some remaining issues to be addressed before acceptance, as outlined below.I would also concur with reviewer 2 that the SNP positions need to be included in this paper or a citation given to a published source. A preprint on bioRxiv would suffice. However, if a preprint is not yet published, please include the positions here.

We have now included the positions in Supplementary file 6 and cited the file in the Materials and methods as follows:

*“*Genotypes were produced using the mass-spectrometry based platform from Agena (Supplementary file 4 and 6).*”*

Reviewer #1:Thanks to the authors for the welcome additions and responses to reviewer feedback.A reiteration of the substantial concern that publishing SNP positions with genetic sequences is an integral part of the work, especially in a journal dedicated to "improv[ing] research communication through open science and open technology innovation".

We have now included the positions in Supplementary file 6.

Reviewer #2:I am happy that the new version does a good job of addressing the initial concerns, and I appreciate the work done by the authors.My only remaining concern is with the application of Bayes rule to obtain Prob (distance | similarity). First, I agree with the choice to swap Figures 3 and Figure 3—figure supplement 1. But the main text reads that there was "no prior" on geographic distance – there was a prior, it was uniform as stated in the Materials and methods section. It is impossible to apply Bayes rule without the use of a prior, and a uniform prior does not represent no information, it actually represents quite specific information. If the intention is to let the genetic data "speak" and not worry about the prior probability of edges being a certain distance then perhaps it would be better to report this as a raw likelihood, i.e. Prob (similarity | distance). Alternatively if the authors really want to stick with the inverted probability then this could be smoothed over with a statement to the effect of "a uniform prior allows the genetic data alone to speak, and more realistic priors can be incorporated by weighting based on the known sampling distances". At the moment it is unfortunately misleading, because the probability of a randomly chosen pair of samples being a given distance apart based on their genetic similarity almost certainly is not given by Figure 3—figure supplement 1.

We have changed “no prior” to “uniform prior” in the main text and added the sentence the reviewer suggested in the Materials and methods to make our statement clear as follows:

“Here, a uniform prior allows the genetic data alone to speak; depending on the question of interest, more realistic priors can be incorporated by weighting based on the known sampling distances.”